# Are EEG Sequences Time Series?
# EEG Classification with Time Series Models and Joint Subject Training

## Abstract

As with most other data domains, EEG data analysis relies on rich domain-specific preprocessing. Beyond such preprocessing, machine learners would hope to deal with such data as with any other time series data. For EEG classification many models have been developed with layer types and architectures we typically do not see in time series classification. Furthermore, typically separate models for each individual subject are learned, not one model for all of them. In this paper, we systematically study the differences between EEG classification models and generic time series classification models. We describe three different model setups to deal with EEG data from different subjects, namely subject-specific models (most EEG literature), subject-agnostic models and subject-conditional models. In experiments on three datasets, we demonstrate that off-the-shelf time series classification models trained per subject perform close to EEG classification models, but that do not quite reach the performance of domain-specific modeling. Additionally, we combine time-series models with subject embeddings to train one joint subject-conditional classifier on all subjects. The resulting models are competitive with dedicated EEG models in 2 out of 3 datasets, even outperforming all EEG methods on one of them.

## 1 Introduction

An electroencephalogram (EEG) measures electrical activity in the brain for diagnostic purposes. EEG classification deals with mapping measured EEG signals to a downstream classification task, usually in the medical domain. This includes for example the identification of sleep stages to identify disorders (Teplan et al., 2002; Subha et al., 2010). While the usage of EEGs has potential for empowering disabled individuals (Al-Qaysi et al., 2018; Wang et al., 2016; Chen et al., 2015) and facilitating stroke rehabilitation (Ang et al., 2015; Alonso-Valerdi et al., 2015), the technological advancement is hindered by substantial obstacles in the consistency and clarity of EEG signals. EEG classification is particularly challenging because EEG signals inherently have a low signal-to-noise ratio (SNR) (Johnson, 2006) and are highly non-Gaussian, non-stationary, and nonlinear (Subha et al., 2010). One promising approach to analyzing EEG sequences is the use of machine learning techniques, such as deep learning, which can adeptly navigate the complexities of this data modality.

The ongoing pursuit for EEG classification has turned towards leveraging sophisticated computational strategies, embodying progressions in deep learning (DL) (Roy et al., 2019) and geometric learning (GL) (Lotte et al., 2007). This includes dedicated manifold attention networks that employ non-Euclidean geometries for incorporating spatial and temporal domains. The research landscape for deep learning-based EEG classification however is isolated, with a lot of methods being developed for this special task. The question of whether EEG classification is closely connected to other areas of deep learning such as time-series classification is rarely investigated. Thus, there is only a small amount of literature that employs established methods and evaluation protocols from other learning domains to EEG classification. While some recent time series literature has done evaluation on EEG data in (Dong et al., 2024; Zhang et al., 2022) the inverse for the EEG domain rarely is the case and only building blocks are adapted. Additionally, the prevalent practice

of training individual models for each subject imposes significant limitations on the scalability, adaptability, and efficacy of EEG classification. These issues arise because the trained model cannot be generalize to other subjects and is unable to reliable predict for unseen patients.

This paper proposes a paradigm shift towards using out-of-the-box time-series classification (TSC) models for EEG classification, challenging existing specialized frameworks, and promoting training a unified model instead of one model per subject. The differences in each of the subjects' brain activity, traditionally a challenge for aggregate training methodologies, are addressed through the use of distinct subject embeddings. To utilize this meta-information about the subject, we present three approaches to incorporate a dedicated subject embedding into a time series classification pipeline for EEG classification. This strategy focuses on the variability between subjects to the model. Our results indicate that established time-series classification approaches with subject embeddings can outperform dedicated state-of-the-art EEG classification models for the task of learning one classification model for all subjects.

This paper presents a step towards embedding EEG classification into well-researched deep learning areas such as time-series classification. Thus it opens the door towards more efficient and better-understood learning on EEG data. Our study presents three primary contributions to the EEG and TSC research landscape:

1. We argue that EEG-series sequences are a special form of time-series, namely time-series with one categorical attribute. This indicates that they should be tackled with time-series models which can incorporate such additional attributes. (Section 3)

2. We provide a theoretical framework to classify EEG models into three categories: (Section 3.4)
   - *subject-specific*, where a separate model is learned for each subject.
   - *subject-agnostic*, where one joint model for all subjects is learned without any subject information.
   - *subject-conditional*, where one joint model for all subjects is learned while utilizing subject information.

3. We propose three novel methodologies for subject-conditional EEG classification. Our procedure is model-agnostic and can be integrated into any differentiable classification model. (Section 4)

4. We show that some simple time-series classification baselines such as ResNet and Inception can be competitive or even outperform dedicated EEG classification models in 2 out of 3 cases, while other TSC models are unable to capture the EEG inherent noise pattern. (Section 6)

Our code can be found in the supplementary materials and will be released upon acceptance.

## 2 Related Work

### 2.1 EEG Classification

In the EEG literature, most models consist of a combination of convolutions (spatial, temporal, and hybrid), normalization, pooling, and a final linear layer. One of the pioneering models is EEGNet (Lawhern et al., 2018), which employs a temporal convolution, two convolutional blocks, and a linear layer. The convolutional blocks comprise depthwise/separable convolutions, batch normalization, ELU activation, and average pooling, followed by dropout.

MAtt (Manifold Attention) (Pan et al., 2022) introduces a novel approach by utilizing a manifold attention layer in the Riemann Space instead of the standard Euclidean Space. It combines a feature extractor, a manifold attention module, and a feature classifier. The feature extractor consists of a 2-layered CNN that convolves over the spatial and spatio-temporal dimensions sequentially. The manifold attention layer transforms the data from Euclidean Space to Riemann Space, applies an attention mechanism using the Log-Euclidean metric, and then converts the data back to Euclidean Space. The feature classifier is a linear layer responsible for class predictions.

Table 1: Comparison of the EEG Classification literature.

| Model | Data Usage | | | | Specialised Layer | Architecture |
| | Single | All | Transfer | Meta | | |
|---|---|---|---|---|---|---|
| MAtt (Pan et al., 2022) | ✓ | – | – | – | Riemann space transformation | Attention |
| MBEEGSE (Altuwaijri et al., 2022) | ✓ | – | – | – | EEGNetBlock | CNN |
| EEG-TCNet (Ingolfsson et al., 2020) | ✓ | – | – | – | TCN-Blocks | CNN |
| TCNet-Fusion (Musallam et al., 2021) | ✓ | – | – | – | notch filter | CNN |
| FBCNet (Mane et al., 2021) | ✓ | – | – | – | spectral filtering | CNN |
| SCCNet (Wei et al., 2019) | ✓ | – | ✓ | – | spatio-temporal filtering | CNN |
| EEGNet (Lawhern et al., 2018) | ✓ | ✓ | – | – | bandpass filtering | CNN |
| Shallow ConvNet (Schirrmeister et al., 2017) | ✓ | – | – | – | spatial filtering | CNN |
| Inception (ours) (Ismail Fawaz et al., 2020) | ✓ | ✓ | – | ✓ | InceptionBlock | CNN |
| ResNet (ours) (Wang et al., 2017) | ✓ | ✓ | – | ✓ | ResNetBlock | CNN |

MBEEGSE (Altuwaijri et al., 2022) employs three independent sequences of convolutional blocks, each utilizing a combination of EEGblocks (Riyad et al., 2020) and SE attention blocks (Altuwaijri & Muhammad, 2022). The outputs of these sequences are concatenated and passed through a fully connected layer. This design allows for the exploration of different kernel sizes, dropout rates, number of temporal filters, and reduction ratios in each sequence, resembling a mixture of experts.

EEG-TCNet (Ingolfsson et al., 2020) comprises three main parts: processing the input through temporal, depth-wise, and separable convolutions; extracting additional temporal features using causal convolutions with batch normalization, dropout, ELU activation, and skip connections; and utilizing a fully connected layer for classification. TCNet-Fusion (Musallam et al., 2021) retains this architecture but concatenates the outputs of the first and second layers before the final classification step.

FBCNet (Mane et al., 2021) follows a similar approach to EEG-TCNet but incorporates spectral filtering in the initial stage. Multiple bandpass filters with varying cut-off frequencies are applied for spectral filtering.

SCCNet (Wei et al., 2019) is a straightforward architecture comprising spatial and spatiotemporal convolutions, a dropout layer, average pooling, and a linear layer for logit generation. The spatiotemporal convolution aims to learn spectral filtering.

ShallowConvNet (Schirrmeister et al., 2017) performs temporal and spatial convolutions, followed by average pooling and a linear layer. The activation function used after convolutions is typically ReLU (Rectified Linear Unit).

In Table 1, we present the architectures used in EEG classification literature along with the corresponding training protocols. Here, *Single* denotes the case where the model solely utilizes data from each subject individually, *All* signifies the usage of data from all subjects without any metadata (for example using the subject Id has an extra feature), *Transfer* indicates the utilization of data from all subjects in a transfer learning setting, and *Meta* implies the utilization of data from all subjects along with the metadata. To the best of our knowledge, we are the first to evaluate the *Meta* setting by incorporating subject information.

Furthermore, we show the specialized layers used in each of the respective models. While earlier architectures like Schirrmeister et al. (2017); Lawhern et al. (2018); Wei et al. (2019); Mane et al. (2021); Musallam et al. (2021) focus on filtering approaches, later works such as Ingolfsson et al. (2020); Altuwaijri et al. (2022) developed specialized building blocks for EEG encoding. MAtt Pan et al. (2022) is the first to successfully incorporate a Transformer architecture by transferring the data from the Euclidean space to a Riemannian manifold.

## 2.2 Time Series Classification

In the time series domain, strategies from other domains, particularly architectures from computer vision, have been successfully applied (Wei et al., 2019; Kachuee et al., 2018a). CNN-based models such as ResNet (He et al., 2016; Wang et al., 2017) or Inception (Szegedy et al., 2015; Ismail Fawaz et al., 2020) have been adapted by applying convolutions over time. Additionally, transformers or attention-based models (Vaswani, 2017) have been tailored to the time series domain by effectively tokenizing inputs before processing them (Kachuee et al., 2018b). Similar to the computer vision domain (Dosovitskiy et al., 2020), attention-based models are considered more effective for high data regimes compared to CNNs, as they provide a global view of the data instead of a local focus imposed by convolutional kernels.

In addition to these adaptations, there are also architectures specifically designed for time series tasks, such as ROCKET (Dempster et al., 2020). ROCKET simplifies time series models while maintaining performance by employing a convolutional layer with multiple randomly initialized kernels of different sizes, dilations, and paddings. The output is then passed to a logistic or ridge regression model. Further notable models include Wu et al. (2023) which is achieving state-of-the-art performance for many TSC problems by stacking TimesBlocks in a residual manner. In Zerveas et al. (2021) a transformer architecture in combination with self-supervised pertaining is used in both time series classification and regression.

There are also initial efforts to integrate EEG datasets into the time series literature. In Dong et al. (2024), the TSLD dataset, which consists of multiple time series datasets from different domains including EEG data, is used for evaluation. Additionally, Zhang et al. (2022) studies the impact of pretraining on the SleepEEG dataset Kemp et al. (2000) for other downstream EEG classification tasks. However, both works focus on the pretraining aspect, comparing only against other methods in this domain, and do not include any dedicated EEG classification models.

## 2.3 Static and time-independent features

In integrating static, time-independent features such as subject IDs into time series data, approaches similar to those in the literature have been identified Leontjeva & Kuzovkin (2016); Tayal et al. (2022). These approaches involve either copying static features for each time step to construct a new time series with additional channels or concatenating static features later in the model with features inferred solely from the time series, typically leading to a model with two separate encoders (one for the static features and one for the time series). This integration of static and time-independent information, such as subjects in EEG datasets, shares similarities with techniques used in the recommender systems literature. In recommender systems, item IDs are utilized to explicitly capture differences between items while employing joint training protocols. Neural networks have been extensively used to embed item information Song & Chai (2018); He et al. (2017), leading to state-of-the-art performance in the field Rashed et al. (2022).

## 3 Understanding EEG data as Time Series with Static Attributes

**Notation:** By $1{:}N := \{1, \ldots, N\}$ we denote the set of the first $N$ integers. By $\mathbb{R}^{*\times C} := (\mathbb{R}^C)^* := \bigcup_{T \in \mathbb{N}} \mathbb{R}^{T \times C}$ we denote the finite sequences of vectors with $C$ dimensions.

### 3.1 Problem Setting

We represent a regularly sampled time series with static attributes by elements $x = (x^{\mathrm{sta}}, x^{\mathrm{dyn}}) \in \mathcal{X} := \mathbb{R}^M \times \mathbb{R}^{*\times C}$, where $x^{\mathrm{dyn}}_{t,c}$ denotes the observed value of channel $c$ at time $t$ and $x^{\mathrm{sta}}_m$ denotes the $m$-th static attribute (not changing over time; $t \in 1{:}\mathrm{len}(x), c \in 1{:}C, m \in 1{:}M$ and $\mathrm{len}(x)$ denotes the length of the time series).

### 3.2 Time Series Classification

Given a sequence of labeled time series data $(x^n, y^n)_{n \in 1:N}$ sampled from an unknown distribution $p$ on $\mathcal{X} \times 1{:}Y$, with $\mathcal{X} \in \mathbb{R}^{T \times C}$ and $Y \in \mathbb{N}$ the number of classes, and a loss $\ell : 1{:}Y \times 1{:}Y \to \mathbb{R}$, the task of **time**

**series classification** then is to find a function $\hat{y} : \mathcal{X} \to 1{:}Y$ (called model) with minimal expected loss

$$\mathbb{E}_{(x,y) \sim p} \; \ell(y, \hat{y}(x)).$$

### 3.3 EEG Classification

EEG classification aims to classify EEG recordings that consist of $C$ sensor readings regularly sampled, e.g., at a set frequency in Hz, usually for a short time range of a few seconds. For each recording additional information, often called meta data, is available, e.g., the ID of a human subject, the sequence number of a session or the sequence number of a task within a session. While the EEG recording is represented by the dynamic part $x^{\mathrm{dyn}}$ of a time series, the additional information fits nicely into static attributes $x^{\mathrm{sta}}$. Thus, from the perspective of the problem setting, EEG classification is just time series classification with static attributes. Furthermore, if we only are interested in the question, to which class a specific EEG recording belongs, we arrive in a special and simple case of time series classification with static attributes. Namely, the case where we only have one static attribute, i.e., $M = 1$. Furthermore, for an EEG dataset $\mathcal{D} \subseteq \mathcal{X}$, we assume that we only have a finite set of layers (the subject IDs), i.e.,

$$\{x^{\mathrm{sta}} \mid x \in \mathcal{D}\} = \{1, ..., S\}.$$

This leads us to the special case under which EEG classification falls, which we call *time series classification with a static categorical attribute*, where our time series has the form

$$x = (x^{\mathrm{sta}}, x^{\mathrm{dyn}}) \in 1{:}S \times \mathbb{R}^{* \times C}.$$

### 3.4 Dealing with Static Attributes in Time Series Classification

In time series classification, static attributes often are represented as constant channels: one adds one channel per static attribute and sets its value to the value of the attribute for all times, i.e.,

$$x^{\mathrm{dyn}}_{t,C+m} := x^{\mathrm{sta}}_m \quad \forall t, m; \; C^{\mathrm{new}} := C + M; M^{\mathrm{new}} := 0. \tag{1}$$

This way, any time series model such as a vanilla convolutional neural network can handle static attributes, even if its input is just a dynamic time series.

However, this approach assumes the static attributes to be numerical and is therefore not dedicated to our task of time series classification with a static categorical attribute. Additionally, in EEG classification signals between subjects often are considered to be so different, that a model is built for each subject individually. To capture the underlying assumption of such an approach, we distinguish three different cases:

**1. The data generating distribution decomposes into subject specific distributions, that share no commonalities:** This means

$$p(y \mid x^{\mathrm{dyn}}, x^{\mathrm{sta}}) = p_{x^{\mathrm{sta}}}(y \mid x^{\mathrm{dyn}})$$

and $p_1, \ldots, p_S$ have nothing in common for the $S$ different subjects. This is the usual EEG classification setting. If this assumption is true, a model per subject can be learned. We call them *subject-specific* models.

**2. The data generating distribution does not depend on the subject ID / static attributes**: Here, we have

$$p(y \mid x^{\mathrm{dyn}}, x^{\mathrm{sta}}) = p(y \mid x^{\mathrm{dyn}}).$$

If this assumption is true, one model for all subjects, that does not have access to the subject ID as input, can be learned. We call such models *subject-agnostic*.

**3. The data generating distribution does depend on the subject ID / static attributes, but does not decompose into completely isolated distributions either**: $p(y \mid x^{\mathrm{dyn}}, x^{\mathrm{sta}})$ depends on both, $x^{\mathrm{dyn}}$ and $x^{\mathrm{sta}}$ in a way that needs to be learned. In this case, a model needs to be learned, that has access to both, the EEG signal and the subject ID. We will propose such *subject-conditional* models in Section 4.

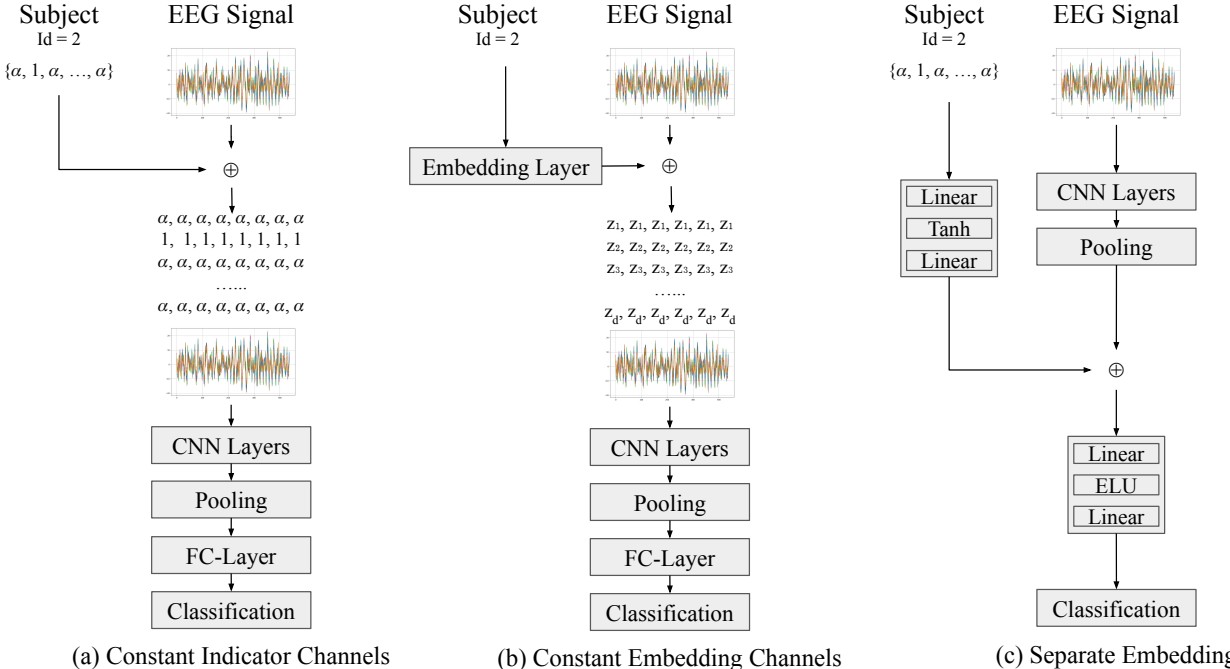

Figure 1: In the figure above we demonstrate the three proposed methods of using the subject information where is (a) Constant Indicator Channels, (b) the Constant Embedding Channels, and (c) Separate Embedding. Here $\oplus$ denotes the concatenation of two tensors across the channel dimension and for (b) $d$ represents the embedding size. We have set the subject Id to 2 as an example.

## 4 Joint Training for Time Series with Static Attributes

If the data-generating distribution decomposes into subject-specific distributions that have no commonalities (i.e., subject-specific), no joint model is needed as we train one model for each occurring value of the static attribute $x^{\mathrm{sta}}$. If the data generating distribution does not depend on the static attribute (i.e., subject-agnostic), it is sufficient to train one time series model that ignores the static attributes.

However, if we assume that the data-generating distribution does depend on the static attributes but does not decompose into completely isolated distributions either, we need to modify existing time-series classification models such that they can incorporate this information properly. Let $\mathcal{D} \subseteq 1{:}S - \mathbb{R}^{*-C}$ be a time series dataset with a categorical attribute. We propose three ways to incorporate the categorical attribute:

**Constant Indicator Channels** The first approach, the Constant Indicator Channels (CIC), introduces a hyperparameter $\alpha \geq 0$ and transforms a time series $(i, X^{\mathrm{dyn}}) \in \mathcal{X}$ with a categorical attribute to a time series with (numerical) static attributes by encoding the categorical attribute via

$$
\begin{aligned}
f_\alpha : 1{:}S &\to \mathbb{R}^S \\
i &\mapsto (\alpha, \ldots, \alpha, \underbrace{1}_{\text{position } i}, \alpha, \ldots, \alpha)
\end{aligned}
\tag{2}
$$

then we can transform that to a time series without static attributes via Equation (1). This transformation can be interpreted as a generalization of one-hot encoding, where higher values of $\alpha$ correspond to higher similarities between different subjects.

**Constant Embedding Channels** For the second approach, Constant Embedding Channels (CEC), we train an embedding layer of dimension $E$ for the subject encoding, represented via a parameterized function

$$
f_\theta : 1{:}S \to \mathbb{R}^E.
$$

We then apply a time series model $\tilde{m} : \mathbb{R}^{T \times (C+E)} \to 1{:}Y$ on the time series which is derived from Equation (1). Thus, if $\phi : \mathbb{R}^E \times \mathbb{R}^{* \times C} \to \mathbb{R}^{* \times (C+E)}$ is the transformation coming from Equation (1), we arrive on the to-trained end-to-end model

$$m : 1{:}S \times \mathbb{R}^{T \times C} \to 1{:}Y$$
$$(i, x^{\mathrm{dyn}}) \mapsto \tilde{m}(\phi(f_\theta(i), x^{\mathrm{dyn}})),$$

where both the parameters of $\tilde{m}$ and the parameters $\theta$ of the subject encoder are trained.

**Separate Embedding**  For the last method we build a Separate Embedding (SE). Let $\tilde{m} : \mathbb{R}^{T \times C} \to 1{:}Y$ be a time-series classification model. Let us furthermore assume, that we can decompose $\tilde{m}$ into a *feature extractor* $f_{\mathrm{FE}} : \mathbb{R}^{T \times C} \to \mathbb{R}^{d_0}$ and a *classification head* $\mathrm{CLF} : \mathbb{R}^{d_0} \to 1{:}Y$ which is given via a multi-layer perception. We again transform subject encodings $i \in 1{:}S$ with the transformation $f_\alpha$ proposed in Equation (2). We then feed this embedding through a multi-layer perceptron $\mathrm{MLP} : \mathbb{R}^S \to \mathbb{R}^{d_1}$. Following the MLP we concatenate the output of this MLP with the output of the feature extractor and apply a classification head $\mathrm{CLF} : \mathbb{R}^{d_0 + d_1} \to 1{:}Y$. The overall model is thus given via:

$$m : 1{:}S \times \mathbb{R}^{T \times C} \to 1{:}Y$$
$$(i, x^{\mathrm{dyn}}) \mapsto \mathrm{CLF}\left( f_{\mathrm{FE}}(x^{\mathrm{dyn}}) \oplus \mathrm{MLP}(f_\alpha(i)) \right),$$

where $\oplus$ denotes the concatenation.

## 5 EEG Classification Models

### 5.1 EEG Classification

For comparison with existing EEG literature, we have opted to expand upon the state-of-the-art MAtt framework (Pan et al., 2022). MAtt represents an advancement in deep learning and geometric learning methodologies, particularly leveraging Riemannian geometry. The model integrates a manifold attention mechanism, aimed at capturing spatio-temporal representations of EEG data on a Riemannian symmetric positive definite (SPD) manifold. By operating within this framework, MAtt capitalizes on the robustness afforded by geometric learning when dealing with EEG data represented in a manifold space. The model is evaluated on three EEG datasets and is found to outperform several leading deep learning methods for general EEG decoding. Additionally, the paper claims that MAtt is capable of identifying informative EEG features and managing the non-stationarity inherent in brain dynamics.

MAtt initiates feature extraction with two convolutional layers, followed by an embedding step transitioning from the Euclidean space to the SPD manifold. The latent representation obtained from the Manifold Attention Module is subsequently projected back into the Euclidean space and channeled into the classification head via a fully connected layer which produces the final prediction.

### 5.2 Time Series Classification

In addition to EEG-specific approaches, we also advocate for the utilization of time series classification models to highlight similarities in EEG datasets with foundational time series data. This approach aims to underscore the generalizability of these models across diverse datasets, establishing them as robust benchmarks and mitigating insular comparisons within the realm of EEG data analysis.

Firstly, we propose the adoption of a standard **ResNet** architecture, renowned for its significant impact in both image processing and time series analysis domains. Our ResNet model comprises three ResNetBlocks followed by the classification head.

Secondly, we selected **Inception** architecture for its multiscale feature extraction capabilities, hierarchical representation learning, and computational efficiency, making it adept at capturing intricate temporal patterns in time series data. For the implementation, we use two different sizes of the model with three and four InceptionBlocks respectively.

Table 2: Summary for the three datasets with the number of subjects, instances, channels, timesteps and classes.

| Dataset | Subjects | Instances | Channels | Timesteps | Classes | Task |
|---------|----------|-----------|----------|-----------|---------|------|
| Mi | 9 | 7.452 | 22 | 438 | 4 | Motor Imagery |
| SSVEP | 11 | 5.500 | 8 | 128 | 5 | Visual Stimulus |
| ERN | 16 | 5.440 | 56 | 160 | 2 | Error recognition |

## 6 Experiments

We compare the time series models ResNet He et al. (2016) and Inception Szegedy et al. (2015) with the previous and current state-of-the-art models for EEG Classification including MAtt Pan et al. (2022), MBEEGSE Altuwaijri et al. (2022), TCNet-Fusion Musallam et al. (2021), EEG-TCNet Ingolfsson et al. (2020), FBC-Net Mane et al. (2021), SCCNet Wei et al. (2019), EEGNet Lawhern et al. (2018), and ShallowConvNet Schirrmeister et al. (2017). For the proposed joint training protocol we evaluate the three proposed methods on ResNet He et al. (2016), Inception Szegedy et al. (2015) and MAtt Pan et al. (2022) as the representative for the EEG models. The results for the other EEG models were taken from Pan et al. (2022), as they have already reproduced all previous work.

### 6.1 Datasets

We conduct our experiments on three common EEG datasets, BCIC-IV-2a **(MI)** (Brunner et al., 2008), MAMEM EEG SSVEP Dataset II **(SSVEP)** (Nikolopoulos, 2021) and the BCI challenge error-related negativity **(ERN)** dataset (Margaux et al., 2012). All preprocessing steps, as well as the train/validation/test splits respectively, follow the same protocol as MAtt (Pan et al., 2022).

**Dataset I - MI** The BCIC-IV-2a dataset is one of the most commonly used public EEG datasets released for the BCI Competition IV in 2008 (Brunner et al., 2008), containing EEG measurements for a motor-imagery task. The EEG signals were recorded by 22 Ag/AgCl EEG electrodes at the central and surrounding regions at a sampling rate of 250 Hz. We applied standard preprocessing procedures to the 22-channel EEG signals. First, we down-sampled the data from 256 Hz to 128 Hz. Next, we applied a band-pass filter to retain frequencies between 4 Hz and 38 Hz. Finally, we segmented the EEG signals into portions starting 0.5 seconds after the onset of the cue and lasting up to 4 seconds, resulting in 438 time points per segment. The motor-imagery task has 4 classes for the imagination of one of four types of movement (right hand, left hand, feet, and tongue). For our experiments, the data is split into 2268/324/2592 instances for train/val/test respectively.

**Dataset II - SSVEP** The MAMEM-SSVEP-II (Nikolopoulos, 2021) contains EEG measurements of an EGI 300 Geodesic EEG System (GES 300). The subjects gazed at one of the five visual stimuli flickering at different frequencies (6.66, 7.50, 8.57, 10.00, and 12.00 Hz) for five seconds. The preprocessing procedures for this dataset included band-pass filtering between 1 Hz and 50 Hz. We then selected eight channels (PO7, PO3, PO, PO4, PO8, O1, Oz, and O2) in the occipital area, where the visual cortex is located. Each trial was segmented into four 1-second segments, starting from 1 second to 5 seconds after the onset of the cue. This resulted in a total of 500 trials of 1-second, 8-channel SSVEP signals for each subject, with each segment consisting of 125 time points. Of the 5 sessions in this dataset, we assigned sessions 1, 2, and 3 as the training set, 4 as the validation set, and 5 as the test set. This split results in 3300/1100/1100 train/val/test instances, respectively.

**Dataset III - ERN** The third dataset BCI Challenge ERN dataset (BCI-ERN) (Margaux et al., 2012) was used for the BCI Challenge on Kaggle[1]. This dataset captures a P300-based BCI spelling task measured by 56 Ag/AgCl EEG electrodes. The spelling task is a binary classification task with an unbalance due to more correct inputs. The preprocessing steps included downsampling from 600 Hz to 128 Hz and applying

---

[1]https://www.kaggle.com/c/inria-bci-challenge

a band-pass filter between 1 Hz and 40 Hz. After preprocessing, each trial consisted of 56 channels and 160 time points. We have split the data into 2880/960/1600 train/val/test instances respectively.

## 6.2 Experimental Setup

We evaluate the time series models ResNet and Inception, as well as the EEG classification architecture MAtt on these three datasets. For the datasets MI and SSVEP, we use Accuracy as the evaluation metric. For ERN we report the Area Under the Curve (AUC) (Rosset, 2004), following the original MAtt paper. We train the models for 500 epochs with early stopping based on validation loss, with a patience of 20 epochs without improvement. For each model, we perform hyperparameter optimization of the batch size $\{32, 64\}$, learning rate $\{1e^{-4}, 1e^{-5}, 1e^{-6}\}$, and weight decay $\{0.0, 0.01, 0.1, 0.5, 1.0\}$. For the subject-conditional task we search for $\alpha$ in $\{0.1, 0.25, 0.5, 0.75\}$. Each set of hyperparameters is evaluated, and the best configuration is selected based on the minimum validation loss averaged across 3 repeats. For the Inception model, we train both variants with three and four InceptionBlocks respectively, and choose the one with the lower validation loss. We then repeat the training 5 times for the best hyperparameter configuration for each subject and in the joint protocol. We report the standard deviation over the runs for each dataset. To evaluate this deviation we use the following equation:

$$\ell^{(r)} := \frac{1}{S} \sum_{i=1}^{S} \ell(y_n, \hat{y}^{(r)}(x^i))$$

$$\text{mean}^{\text{runs}} := \frac{1}{R} \sum_{r=1}^{R} \ell^{(r)}$$

$$\text{stddev}^{\text{runs}} := (\frac{1}{R-1} \sum_{r=1}^{R} (\ell^{(r)} \times \text{mean}^{\text{runs}})^2)^{\frac{1}{2}}$$

Here, $R$ is the total number of runs. For our main baseline model MAtt, we have taken the best hyperparameters reported by the authors. However, we were unable to reproduce the results. In Table 3, we, therefore, report our reproduction of the MAtt as well as the reported results denoted as MAtt[†] as well as common baselines from the EEG literature.

## 6.3 Performance Comparison

In this section, we evaluate the subject-specific, subject-agnostic, and subject-conditional models on the three datasets. Our results are shown in Table 3.

For the *subject-specific* approach, we compare our proposed TSC baseline Inception and ResNet with the leading EEG literature, where MAtt represents our reproduction of the original paper and MAtt[†] denotes the reported results by the authors. The results for the other models were extracted from (Pan et al., 2022). For this approach, the TSC models achieve competitive results for the SSVEP and ERN datasets, where the Inception model in particular produces results close to the state-of-the-art, whereas, ResNet underperforms, especially for the SSVEP dataset which we show in Table 5 and Table 6 respectively. Here we can observe the standard behavior for EEG datasets, where the model performance varies depending on the subject. For the MI dataset, the designated EEG models emerge as clear winners over the time series models as shown in Table 4. Here the patching for the input sequence employed by MAtt proves to be beneficial for this particular dataset with its longer sequence length, while the vanilla Inception and ResNet models have no such functionality.

Secondly, for the *subject-conditional* approach, we evaluate our proposed subject-conditional methods CIC, CEC, and SE for the Inception, ResNet, and MAtt architectures (Table 7), and report the best-performing subject-conditional models, which is selected on validation loss, under MAttJoint, ResNetJoint, and InceptionJoint in Table 3, respectively. With the additional subject information and the ability to learn patterns across the subjects in a joint manner, we beat all other baselines for the SSVEP dataset. Additionally, we produce the second-best results on the ERN dataset. Generally, the Id embedding method in CEC yields

Table 3: Performance comparison for the datasets BCIC-IV-2a (MI), MAMEM EEG SSVEP Dataset II (SSVEP) and the BCI challenge error-related negativity (ERN). We report the average accuracy for MI and SSVEP and the AUC for ERN over 5 runs respectively. The first block contains the common baselines for the EEG literature, where MAtt$^\dagger$ are the results reported in the original paper and MAtt is our reproduction. The second block is composed of the proposed TSC models ResNet and Inception for the subject-specific approach. The last block consists of the subject-conditional approach where all subjects are trained jointly and we utilize the subject information. The best result is highlighted in bold and the second best is underlined.

| Models | MI | SSVEP | ERN |
|---|---|---|---|
| ShallowConvNet | 61.84±6.39 | 56.93±6.97 | 71.86±2.64 |
| EEGNet | 57.43±6.25 | 53.72±7.23 | 74.28±2.47 |
| SCCNet | 71.95±5.05 | 62.11±7.70 | 70.93±2.31 |
| EEG-TCNet | 67.09±4.66 | 55.45±7.66 | **77.05**±2.46 |
| TCNet-Fusion | 56.52±3.07 | 45.00±6.45 | 70.46±2.94 |
| FBCNet | 71.45±4.45 | 53.09±5.67 | 60.47±3.06 |
| MBEEGSE | 64.58±6.07 | 56.45±7.27 | 75.46±2.34 |
| MAtt$^\dagger$ | 74.71±5.01 | 65.50±8.20 | 76.01±2.28 |
| MAtt | **74.37**±3.39 | 63.90±1.95 | 69.28±5.31 |
| ResNet (ours) | 58.05±3.68 | 43.49±3.41 | 69.16±5.16 |
| Inception (ours) | 62.85±3.21 | 62.71±2.95 | 73.55±5.08 |
| MAttJoint (ours) | 61.13±0.56 | 60.71±0.29 | 75.78±1.23 |
| ResNetJoint (ours) | 55.54±2.72 | 54.15±1.19 | 73.09±0.72 |
| InceptionJoint (ours) | 61.38±1.57 | **66.00**±0.36 | 76.13±0.95 |

Table 4: Performance Comparison on the MI dataset for the subject-specific case. We report the average accuracy over 5 runs respectively.

| Subject | Inception | ResNet | MAtt |
|---|---|---|---|
| 1 | 78.96 ± 1.82 | 72.83 ± 3.22 | **86.94** ± 1.36 |
| 2 | 41.25 ± 2.63 | 38.12 ± 3.27 | **56.94** ± 2.42 |
| 3 | 82.09 ± 3.05 | 70.76 ± 3.10 | **88.33** ± 1.17 |
| 4 | 52.43 ± 2.40 | 49.72 ± 3.41 | **67.85** ± 3.42 |
| 5 | 38.75 ± 3.74 | 37.50 ± 4.51 | **61.32** ± 1.07 |
| 6 | 48.61 ± 1.78 | 44.58 ± 4.39 | **52.50** ± 2.52 |
| 7 | 75.99 ± 4.51 | 62.36 ± 3.39 | **91.18** ± 0.89 |
| 8 | 74.31 ± 3.28 | 73.68 ± 2.54 | **83.06** ± 2.01 |
| 9 | 73.26 ± 2.14 | 72.92 ± 2.24 | **81.18** ± 1.06 |
| **Summary** | 62.85±3.21 | 58.05±3.68 | **74.37**±3.39 |

the best outcome and is able to learn better embeddings compared to the weighted one-hot encoding in the CIC and SE approaches. For MI, we observe that the joint training procedure provides no benefits, and the subject-specific models have a clear advantage.

It is important to note that the Inception architecture demonstrates significantly faster computational performance, with a speed-up factor of 11.2 and 9.8 for the three and four-layer variants respectively, while ResNet exhibits a speed-up factor of 10.4 in wall-clock time compared to the MAtt architecture. These measurements were obtained while running the models on the same NVIDIA GeForce RTX 4070 Ti GPU. The main computational cost of MAtt arises from its full Attention mechanism (Vaswani, 2017) compared to the lightweight CNN counterparts.

These findings suggest that time series models are well-suited for EEG classification tasks and can be readily applied for both subject-specific and subject-conditional cases, offering promising performance and computational efficiency.

Table 5: Performance Comparison on the SSVEP dataset for the subject-specific case. We report the average accuracy over 5 runs respectively.

| Subject | Inception | ResNet | MAtt |
|---------|-----------|--------|------|
| 1 | $80.40 \pm 2.06$ | $56.40 \pm 3.98$ | $\mathbf{81.60} \pm 2.87$ |
| 2 | $86.60 \pm 1.62$ | $75.20 \pm 3.82$ | $\mathbf{89.40} \pm 1.36$ |
| 3 | $\mathbf{61.60} \pm 3.07$ | $38.20 \pm 2.64$ | $58.20 \pm 5.64$ |
| 4 | $\mathbf{25.00} \pm 4.00$ | $20.40 \pm 4.41$ | $20.60 \pm 3.88$ |
| 5 | $25.00 \pm 6.72$ | $22.60 \pm 1.74$ | $\mathbf{26.40} \pm 4.80$ |
| 6 | $\mathbf{79.20} \pm 1.72$ | $38.20 \pm 3.92$ | $79.00 \pm 2.68$ |
| 7 | $\mathbf{69.20} \pm 1.72$ | $42.80 \pm 2.14$ | $66.00 \pm 2.19$ |
| 8 | $23.60 \pm 1.74$ | $23.40 \pm 1.85$ | $\mathbf{23.80} \pm 2.71$ |
| 9 | $79.40 \pm 2.58$ | $62.40 \pm 3.20$ | $\mathbf{88.20} \pm 2.04$ |
| 10 | $68.60 \pm 3.72$ | $35.00 \pm 3.03$ | $\mathbf{70.60} \pm 4.54$ |
| 11 | $\mathbf{91.20} \pm 2.48$ | $63.80 \pm 6.79$ | $90.20 \pm 1.47$ |
| Summary | $62.71 \pm 2.95$ | $43.49 \pm 3.41$ | $\mathbf{63.90} \pm 1.95$ |

Table 6: Performance Comparison on the ERN dataset for the subject-specific case. We report the average AUC over 5 runs respectively.

| Subject | Inception | ResNet | MAtt |
|---------|-----------|--------|------|
| 2 | $81.14 \pm 4.60$ | $74.03 \pm 3.37$ | $\mathbf{81.86} \pm 2.56$ |
| 6 | $\mathbf{90.89} \pm 2.02$ | $88.40 \pm 2.67$ | $68.34 \pm 1.76$ |
| 7 | $83.45 \pm 6.80$ | $\mathbf{85.45} \pm 7.05$ | $69.28 \pm 10.90$ |
| 11 | $59.50 \pm 10.52$ | $55.27 \pm 9.19$ | $\mathbf{70.50} \pm 2.66$ |
| 12 | $\mathbf{74.49} \pm 3.67$ | $68.62 \pm 9.92$ | $60.62 \pm 4.70$ |
| 13 | $56.34 \pm 2.26$ | $51.15 \pm 4.16$ | $\mathbf{62.92} \pm 3.65$ |
| 14 | $\mathbf{78.96} \pm 2.47$ | $75.22 \pm 3.17$ | $70.42 \pm 4.80$ |
| 16 | $\mathbf{56.09} \pm 5.53$ | $50.53 \pm 3.54$ | $55.71 \pm 2.56$ |
| 17 | $79.93 \pm 3.78$ | $75.93 \pm 4.20$ | $\mathbf{80.60} \pm 5.52$ |
| 18 | $\mathbf{77.40} \pm 1.73$ | $72.30 \pm 1.39$ | $75.11 \pm 1.42$ |
| 20 | $\mathbf{65.90} \pm 4.96$ | $59.97 \pm 2.56$ | $57.60 \pm 8.12$ |
| 21 | $\mathbf{74.53} \pm 6.72$ | $67.95 \pm 7.76$ | $62.43 \pm 9.56$ |
| 22 | $\mathbf{95.87} \pm 2.03$ | $95.09 \pm 0.89$ | $89.33 \pm 4.28$ |
| 23 | $69.85 \pm 7.00$ | $61.66 \pm 2.38$ | $\mathbf{70.03} \pm 5.40$ |
| 24 | $\mathbf{77.66} \pm 1.36$ | $70.88 \pm 4.60$ | $73.71 \pm 2.99$ |
| 26 | $54.90 \pm 5.80$ | $54.03 \pm 4.66$ | $\mathbf{60.08} \pm 1.92$ |
| Summary | $\mathbf{73.55} \pm 5.08$ | $69.16 \pm 5.16$ | $69.28 \pm 5.31$ |

## 6.4 Ablation Study

As an ablation study for the *subject-agnostic* case, we also evaluate joint training without any subject information added. We refer to this method as SA in Table 7, where in 7 out of 9 cases, the proposed methods of utilizing subject information outperform the model variant without it. Hence, incorporating IDs into the learning process via static features generally enhances performance. Note, that most models only witness a small performance drop when being trained in a subject-agnostic manner. It especially turns out, that a subject-agnostic vanilla Inception beats the current state-of-the-art on **SSVEP**. This is another strong argument for considering time-series classification models as baselines in EEG classification. In addition to the better performance, the standard deviation of all *subject-conditional* models is much lower compared to its *subject-specific* counterparts demonstrating better generalization.

Table 7: Performance comparison with subject information for the joint training protocol for the Constant Indicator Channels (CIC), Constant Embedding Channels (CEC), and Separate Embedding (SE). (SA) is the subject-agnostic approach with no additional subject information. We report the average accuracy for MI and SSVEP and the AUC for ERN over 5 runs respectively. The best result per dataset is marked in bold and the best method per model underlined.

| Method | MI | SSVEP | ERN |
|--------|------|--------|------|
| **ResNet** | | | |
| SA | 55.42±2.72 | 50.48±0.48 | 69.76±0.72 |
| CIC | 55.54±1.72 | 50.39±0.24 | 73.09±0.66 |
| CEC | 55.21±2.52 | 54.15±1.19 | 73.06±2.21 |
| SE | 49.08±2.07 | 49.88±0.75 | 68.77±1.43 |
| **Inception** | | | |
| SA | 59.21±1.39 | 59.73±1.35 | 75.15±1.16 |
| CIC | 58.55±2.00 | **66.00**±0.36 | 75.28±0.53 |
| CEC | **61.38**±1.57 | 65.79±0.87 | **76.13**±0.95 |
| SE | 54.80±3.01 | 64.00±0.36 | 74.29±1.38 |
| **MAtt** | | | |
| SA | 61.13±0.56 | 60.71±0.29 | 75.77±0.72 |
| CIC | 60.56±0.20 | 59.80±0.86 | 75.02±1.14 |
| CEC | 60.14±0.94 | 60.20±0.83 | 75.78±1.23 |
| SE | 60.31±1.47 | 60.23±0.74 | 73.39±0.54 |

## 7 Conclusion and Future Work

In this paper, we bridged the gap between time series analysis and EEG processing. We argued that EEG data can be seen as time-series data with static attributes. Furthermore, we showed that established models for time-series classification can be competitive with methods specifically dedicated to EEG classification. While EEG classification is usually evaluated by training an individual model per subject, we also train joint time-series classification models for all subjects. By incorporating subject embeddings into the classification process, we showed that these subject-conditional time series models can be competitive or even outperform dedicated EEG approaches which are trained for all subjects individually.

We see our contribution as a first step towards integrating recent advancements in deep learning methodology into the field of EEG classification. While this paper demonstrates that well-established time-series baselines should be considered for EEG data, not all TSC models can adapt to the EEG-specific noise patterns. By opening up isolated research in EEG classification to more general learning domains, a natural future direction is to explore whether recent trends in machine learning should be transferred to EEG analysis. This could include, for example, the development of foundation models for EEG data. Several open issues need to be addressed, such as possible pre-training objectives, evaluation scenarios, and tasks specific to EEG analysis. Additionally, an in-depth analysis of EEG-specific noise patterns and how to encode this data modality is required in future work.

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
