# OpenReview forum: "Are EEG Sequences Time Series? EEG Classification with Time Series Models and Joint Subject Training"
_TMLR — Rejected by TMLR_

### Review · Reviewer_QAxL · 2024-08-01

**Summary Of Contributions:**

This paper compares non specialized time-series models (ResNet and Inception) for EEG data to specialized models for EEG data. They find that the nonspecialized models are competitive, but not state of the art in most cases.

They propose 3 ways to formulate EEG as a time series classification: subject-specific, subject-agnostic, and subject conditional.

**Audience:**

Yes

**Claims And Evidence:**

No

**Requested Changes:**

## Structure of claim
_overall: advantage of generic models needs to be explicit_

- end of page1, para1, it sys there is a "pressing need for DL algorithms" ... there is no support that the solution *must* be deep learning, the evidence supports there is room for improvement. There is no mention of downstream tasks or the real world impact of tasks so, the need for improvement is not even explicit
- p1, para2 says there is little adoption of evaluation from other domains in EEG, this seems appropriate, there is no reason that this is advantageous; evaluation *should* be domain specific for interpretation and action based on the evaluation. either remove this or provide evidence to support that **evaluation** procedures being adopted across domains has a positive impact on the domain knowledge production
- the limitations of learning for each subject are not stated specifically (end of pg1,pa2)

## related works
- citations for resnet and inception in section 5 par 1 is are secondary papers, not the actual papers that present those architectures; this is misleading because it makes them not seem older than the specific architectures

## Experimental setup
- why only replicate MAtt and use their results for all other models? All comparisons should be treated consistently at least, and ideally all should be replicated. If there’s a reason for the chosen setup, it needs to be stated explicitly otherwise other models should also be reimplemented
- in 5.2 par 1 “we repeat the training 5 times on a random seed” is unclear. Does this mean 5 different random seeds, or the same random seed 5 times, which should have no variation?
- reporting both mean and a notion of spread is good, but in table 3 caption what spread metric is reported is not stated, and more importantly, most results are overlapping ranges.  A statistical test to make the comparisons appropriately and clarify what the ranges are would make the results more clear and meaningful
- the test train splits need to be at least better documented and possibly re-run to make the interaction with individual people more clear; the subject-specific models clearly require train/val/split *for each person* and for the agnostic case it’s unclear how individuals interact with the splits; as that would change the interpretation of what is being learned if that’s average performance across individuals without an examination of the per individual performance. Table 4 clearly shows that the individual models vary in performance wildly across individuals, with some individuals having overall worse performance for all models. So, the impact of individual on test performance is an important indicator in making actually usable predictions.

## Language/Writing
these are minor but would improve the paper
- page 2 mixes "subjects" and "patients" seemingly interchangeably; this is inappropriate. "participant" is currently the best term to use to refer to healthy people from whom research data is collected; "subjects" was correct but is considered outdated (though institutionally still used) "patient" is generally only used in medical scenarios where individuals have diagnosed conditions.
- the EEG classification related work reads like a list. It should unify themes across the work and situate them better, like the other 2 parts of related work
- in section 2.2, the use of spatial vs temporal correlations reads inconsistently at times. Eg inception is treated throughout as a generic time series model, but described as having spatial convolutions; this should be explained if correct or fixed if typo
- section 2.3 what group means is unclear
- section 3.0.4 introduces a task of predicting which person a sample belongs to and it’s not clear why that would be useful but that task is also not what is solved so this reads as just a distraction; it should either be better motivated or removed
- in all data descriptions the prediction task is only stated implicitly by describing classes
- description of SSVEP dataset mentions 5 sessions and 5 frequencies; train test split is across sessions, but it is not clear if that means it is also across frequencies or if all frequencies are present in all sessions. If all are present it’s a standard classification task, if they’re divided then this is a weird choice to split the data
- description of the ERN dataset does not use the word “instances” in the train/val/test split like the other two do

**Strengths And Weaknesses:**

## Strengths

- the work is very clear methodologically and has good replicability
- the subject specific/agnostic/conditional formulation is clearly presented and well reasoned and helps connect the literature

## Weaknesses


- the contribution for the work is unclear, the compared baselines draw inspiration from resnet and adapt the basic ideas from resnet to EEG data, several of the original papers cited within Pan et al, cite resnet and some include residual blocks
- it is unclear what the advantage of generic, nonspecialized models is, the work seems to rely on an implicit superiority. This might be clear to readers who are focused in the deep learning literature but is unlikely to make a compelling case to those who work on EEG, which seems to be the target reader, so this should be made more clear, what gains there are to using generic architectures for EEG specifically.  "efficiency" is stated, but that is non specific, is it time efficienty in training, memory efficiency, time efficiency at inference? all of the above? overall this point needs to be more specific and explicit in order to meet the criterion of well supported claims.
- the conceptual comparison of the EEG specific models to the proposal in table 1 is inadequately vague. more clear, explicit analytical comparison of resnet to these other models would make the argument stronger of what to gain from the generic classifiers or provide insight into what the specialized classifiers might be missing. For example EEG-TCNet includes residual blocks and TCNet fusion combines that with EEG-net; this is not clear in the comparison and since they have residual blocks
- the generic models weakly outperform the specialized models in a small number of cases, so the conclusions that generalized models should be considered more strongly is also a weak argument
- the conclusion recommends considering attention-based models as future work, but the slow computation time of MAtt is blamed on its attention components in section 5.3 (2nd to last paragraph) this is minor but undermines some of the argument structure
- at least one paper using resnet on eeg data (albiet for a different task) exists, so this should be reconciled. [Xuanjie Qiu, Fang Yan,
 Haihong Liu, "A difference attention ResNet-LSTM network for epileptic seizure detection using EEG signal, Biomedical Signal Processing and Control" https://doi.org/10.1016/j.bspc.2023.104652.]

---

> ### Author Response · Authors · 2024-09-09
> **Response to Reviewer QAxL**
>
> We thank the reviewer for their detailed review and would like to address the individual points:
>
> **Structure of claim**
>
> Advantages of generic models
> ```
> Generic models have no inherent advantage over specialized models; however, we show that these basic models perform similarly
> to or even better than specialized models. Therefore, they should be incorporated into EEG classification research, which
> is currently quite disconnected from the broader time series literature. In the specific case of this paper, in addition
> to the performance increase, there is also a drastic reduction in runtime as demonstrated in Section 6.3. paragraph 4 by a factor of 9-11.
> ```
>
> Need for DL algorithms
> ```
> We have revised this paragraph to better reflect the range of other model architectures as well.
> ```
>
> Adoption of other domains in EEG
> ```
> We agree that the evaluation should be domain-specific; that's why we evaluate all models on EEG datasets. However, we
> have shown that a vast array of time series models has yet to be applied to the field of EEG classification and that
> these models can be used to improve the performance of EEG classification. We have clarified the wording in that paragraph.
> ```
>
> limitations of Subject-specific models
> ```
> The limitation is that the resulting model is incapable of generalizing to new subjects, which makes it less useful for
> real-world applications. We have clarified this point in the paper.
> ```
>
> citations for resnet and inception
> ```
> Thank you for pointing this out. We have updated the citation to refer to the original papers.
> ```
>
> **Experimental setup**
>
> Replication MAtt
> ```
> MAtt has already reproduced the results for all other models, and these results match those from the original papers.
> Therefore, we have chosen to use their results for all other models and focus solely on reproducing MAtt itself.
> We added this justification to the paper in the first paragraph of section 5.
> ```
>
> Random seeds
> ```
> At the start of each training run, we generate a new dataloader using a different random seed. We have clarified this process
> in the paper.
> ```
>
> Table 3 std
> ```
> Table 3 follows the same calculation as described in the stated equation. We agree that the high standard deviations for
> the subject-specific models in Table 3 are problematic and make it difficult to compare different models, as most of the
> approaches fall within one standard deviation. This large standard deviation is due to the high variance in results for
> each subject in the subject-specific case. Joint training approaches have a clear advantage here because they generalize
> better across subjects, resulting in lower variance in the results. We have added this explanation to the paper in the end
> of section 5.4.
> ```
>
> test/train splits
> ```
> Each subject is represented in the validation and test splits with the same number of samples. We have clarified this in the paper.
> ```
>
> **Language/Writing**
>
> Page 2 mixes "subjects" and "patients" seemingly interchangeably
> ```
> We have replaced all mentions of patients and changed them to subjects to be consistent with the literature.
> ```
>
> the EEG classification-related work reads like a list. It should unify themes across the work and situate them better,
> like the other 2 parts of related work
> ```
> We have added a paragraph at the end of Section 2.1 to summarize the related work with Table 1 and illustrate the development over time.
> ```
>
> in section 2.2, the use of spatial vs temporal correlations reads inconsistently at times. Eg inception is treated
> throughout as a generic time series model, but described as having spatial convolutions
> ```
> We have clarified this by replacing 'spatial' with 'applying convolutions over time'
> ```
>
> section 2.3 what group means is unclear
> ```
> The group formulation was indeed unclear and we have removed it.
> ```
>
> section 3.0.4 task introduction
> ```
> There was a mixup in the formulation. We have changed it to:
> 'Furthermore, if we only are interested in the question, to which class a specific EEG recording
> belongs, we arrive in a special and simple case of time series classification with static attributes.'
> ```
>
> Dataset descriptions
> ```
> We have added the specific task to Table 2 as an additional task column. Additionally, for the SSVEP we have added the
> word instances. All frequencies occur in a given session for this dataset.
> ```

---

### Review · Reviewer_jvtZ · 2024-08-27

**Summary Of Contributions:**

The paper presents a study of "time series classification" algorithms on EEG signals, comparing several baseline methods to the authors' proposed methods on three pre-existing datasets (and tasks). Authors study three different settings: subject-specific, subject-agnostic, and subject-conditional, which differ in the way in which the "subject" information is being used, since EEG signals can differ across subjects. Authors implement neural network based on ResNet, Inception and MAtt (Manifold Attention) architecture, and show that their implementation delivers in general close to state-of-the-art results.

**Audience:**

Yes

**Claims And Evidence:**

No

**Requested Changes:**

- Authors should provide a clear definition of "time series models" -- IIUC the models used here classify temporally ordered data points that are all presented at once, and with constant, known length.
- Is "session dependence" (in addition to subject dependence) a problem in these EEG tasks?
- Please include a discussion of speech recognition in the related work section, or explain why this is not relevant. Speech recognition has speaker dependence and many approaches have been published for "speaker adaptive" speech recognition or more general "contextualized ASR" recently
- Include a discussion of the signal pre-processing that was used in this work
- Explain what attempts were made to resolve the non-reproducibility of results and how significant are the differences reported in the paper

**Strengths And Weaknesses:**

Strengths
----

- The paper treats an interesting and relevant area; EEG signals and medical domain in general can benefit from deep learning approaches and deserve further study
- The paper is generally well written and easy to understand
- The authors present a comprehensive set of carefully done experiments that they compare to a baseline (from (Pan et al., 2022))

Weaknesses
----

- The paper lacks a proper definition of "time series models" and has little focus -- "joint subject training" is interesting in itself, but I am surprised that authors do not discuss speech recognition models (a classic "time series" problem) or financial data
- The paper does not discuss data pre-processing in any depth, which presumably has a significant influence as well
- Significance and reproducibility of results seem to be a problem, error bars are quite wide and results seem to be difficult to reproduce (authors deserve credit for being transparent here, though)

---

> ### Author Response · Authors · 2024-09-09
> **Response to Reviewer jvtZ**
>
> We thank the reviewer for their detailed review and would like to address the individual points:
>
> Authors should provide a clear definition of "time series models":
>
> Yes, that understanding is correct. We have clarified this in the paper by defining $X$ by $ R^{ T \times C}$ (section 3.3)
>
>
> Is "session dependence" (in addition to subject dependence) a problem in these EEG tasks?
> ```
> Yes, session dependence can also be a problem and introduces a slight covariate shift. However, finding common patterns
> across subjects is the more challenging problem to solve, which is why it is the focus of our paper.
> ```
>
> Please include a discussion of speech recognition in the related work section, or explain why this is not relevant.
> Speech recognition has speaker dependence and many approaches have been published for "speaker adaptive" speech recognition
> or more general "contextualized ASR" recently
> ```
> The difference between EEG and speech recognition is that EEG signals suffer from a lower signal-to-noise ratio, with
> noise varying more significantly between individuals compared to speech recognition. This is why joint model training is common in speech recognition but less prevalent in EEG classification.
>
> ```
> Include a discussion of the signal pre-processing that was used in this work
> ```
> As mentioned in the response to reviewer RCq4 we have added the preprocessing steps to section 5.1 in the dataset
> descriptions.
>
> This is the preprocessing done by MAtt [1]:
> Data set I (MI)
> We performed standard preprocessing procedures for the 22-channel EEG signals, including 1)
> Down-sampling from 256 Hz to 128 Hz, 2) Band-pass filtering at 4-38 Hz, and 3) Segmenting EEG
> signals at 0.5-4s (438 timepoints) of the onset of cue for each trial
>
>
> Data set II (SSVEP)
> The preprocessing procedures for this dataset
> were 1) Band-pass filtering at 1-50 Hz, 2) Selecting eight channels (PO7, PO3, PO, PO4, PO8, O1,
> Oz, and O2) in the occipital area, the location of visual cortex, and 3) Segmenting each trial into four
> 1-second segments at 1s-5s of the onset of cue, yielding a total of 500 trials of 1-second 8-channel
> SSVEP signals for each subject, thus the time length of input EEG data is 125
>
> Data set III (ERN)
> The preprocessing steps include 1) Downsampling from 600 Hz to 128 Hz and 2) Band-pass filtering at
> 1-40 Hz. Each trial has a size of 56 channels by 160 timepoints after the preprocessing step.
>
> [1] Yue-Ting Pan, Jing-Lun Chou, and Chun-Shu Wei. Matt: A manifold attention network for eeg decoding.
> Advances in Neural Information Processing Systems, 35:31116–31129, 2022.
> ```
>
> Explain what attempts were made to resolve the non-reproducibility of results and how significant are the differences
> reported in the paper
> ```
> Upon contacting the authors, we re-ran the experiments using both the hyperparameter selections they provided and our
> reported search space. The differences are unfortunately significant, as shown in Table 3; however, they remain within
> the standard deviation. Overall, a high standard deviation is common in EEG literature due to the low signal-to-noise
> ratio of the measurements. Additionally, we would like to point out that our joint training approach outperforms both
> our reproduction of the paper and the results reported in the original paper. Moreover, all our experiments are
> reproducible with the code and hyperparameter selections we have published.
> ```

---

### Review · Reviewer_RCq4 · 2024-08-27

**Summary Of Contributions:**

This paper investigates the classification of EEG data and proposes to treat EEG as a time series with static attributes. Extensive experiments are provided to verify that current time series classification models can perform close to EEG classification models. A joint subject-conditional classifier is presented, which can beat individually trained models in 2 of 3 datasets.

**Audience:**

Yes

**Claims And Evidence:**

No

**Requested Changes:**

The authors should clarify the "gap" or unique difference between EEG and time series and compare with the above-mentioned baselines.

**Strengths And Weaknesses:**

## Strengths

-	This paper is well-written and clear.
-	Extensive experiments and baselines are provided.
-	The joint-training model results are interesting and demonstrate the possibility of building a unified model for all subjects.

## Weaknesses
1.	The main idea of this paper is kind of meaningless.
Since this paper studies how to bridge time series with EEG classification, a foundational idea is that EEG has a clear gap w.r.t. canonical time series. I do not think this idea is well supported given that a lot of time series papers have already experimented with EEG data, such as [1][ 2]

[1] TimeSiam: A Pre-Training Framework for Siamese Time-Series Modeling, ICML 2024

[2] Self-Supervised Contrastive Pre-Training For Time Series via Time-Frequency Consistency, NeurIPS 2022

Thus, I think the “gap” between time series and EEG is not as large as the authors assumed. This problem will seriously affect the contribution of this paper.

2.	Some advanced baselines are missing.

There are several latest time-series classification models that should be considered.

[1] TimesNet: Temporal 2D-Variation Modeling for General Time Series Analysis, ICLR 2023

[2] A Transformer-based Framework for Multivariate Time Series Representation Learning, SIGKDD 2021

---

> ### Author Response · Authors · 2024-09-09
> **Response to Reviewer RCq4, Part 1**
>
> We thank the reviewer for their detailed review and would like to address the individual points:
>
> Comparison to published work
> ```
> Thank you for pointing out these papers; they look very interesting, and we have included them in our related work section.
> We were not aware of [1] because it was published after our submission; however, we missed [2]. After reading both papers,
> we found that they focus on the pretraining aspect, while our main focus is on bridging the gap between the time series
> and EEG literature, as well as joint training across subjects. Furthermore, both mentioned papers do not account for any
> EEG-specific models as baselines and only compare different preprocessing methods. We agree with the reviewer that EEG
> data is often considered "normal" time-series data in the time-series community. However, on the other hand, the EEG
> community (e.g., FBCNet, MBEEGSE, MAtt) does not take general time-series classification models into account. As our
> experimental evaluation shows, basic time series models can already be competitive and should be considered by the EEG
> community. Furthermore, the vanilla time-series literature usually does not consider static information such as subject
> IDs. Our work builds a bridge by proposing embedding techniques for subject-conditional models. We have clarified this
> in our revision.
> ```

---

> ### Author Response · Authors · 2024-09-09
> **Response to Reviewer RCq4, Part 2**
>
> Additional baselines:
> ```
> We have implemented TimesNet, and it is working great in the joint setting for ERN. However, it is not performing well
> at all for the other two datasets, producing near-random performance. We conducted extensive hyperparameter tuning
> both in their framework and in our own to ensure there were no bugs, however, these models are unable to capture the
> EEG noise patterns. We hypothesize that TimesNet has an overly parameterized classification head, which does not
> perform well for small EEG datasets. We also looked at the second paper and found ConvTran in [1], which outperforms
> it, so we decided to implement ConvTran as an additional baseline. Here, we tuned for the number of heads, the
> internal dimension, and the fixed positional encoding, as well as the other hyperparameters we reported in the paper.
> Due to time constraints, we repeated the experiments 2 times instead of 5 times, compared to the other models in the
> paper.
>
> [1] Foumani, Navid Mohammadi, Chang Wei Tan, Geoffrey I. Webb, and Mahsa Salehi. “Improving Position Encoding of
> Transformers for Multivariate Time Series Classification.” Data Min. Knowl. Discov. 38, no. 1 (2024): 22–48.
>
> Here are the current results for this model:
>
> ```
>
> MI
> | Subject   | ConvTran            |
> |-----------|-----------------------|
> | Subject 1 | 0.6458 $\pm$ 0.0139   |
> | Subject 2 | 0.3056 $\pm$ 0.0556   |
> | Subject 3 | 0.7188 $\pm$ 0.0312   |
> | Subject 4 | 0.4809 $\pm$ 0.0122   |
> | Subject 5 | 0.2517 $\pm$ 0.0122   |
> | Subject 6 | 0.3837 $\pm$ 0.0017   |
> | Subject 7 | 0.6840 $\pm$ 0.0104   |
> | Subject 8 | 0.6667 $\pm$ 0.0069   |
> | Subject 9 | 0.7205 $\pm$ 0.0122   |
> | Summary   | 0.5397                |
>
> SSVEP
> | Subject   | ConvTran            |
> |-----------|-----------------------|
> | Subject 1 | 0.4850 $\pm$ 0.0650   |
> | Subject 2 | 0.2400 $\pm$ 0.0400   |
> | Subject 3 | 0.3250 $\pm$ 0.0050   |
> | Subject 4 | 0.2200 $\pm$ 0.0300   |
> | Subject 5 | 0.2200 $\pm$ 0.0000   |
> | Subject 6 | 0.2850 $\pm$ 0.0250   |
> | Subject 7 | 0.3150 $\pm$ 0.0150   |
> | Subject 8 | 0.2200 $\pm$ 0.0200   |
> | Subject 9 | 0.4650 $\pm$ 0.0250   |
> | Subject 10| 0.1900 $\pm$ 0.0100   |
> | Subject 11| 0.3300 $\pm$ 0.1000   |
> | Summary   | 0.2995                |
>
> ERN
> | Subject   | convtran           |
> |-----------|-----------------------|
> | Subject 2 | 0.7259 $\pm$ 0.0099   |
> | Subject 6 | 0.7947 $\pm$ 0.0286   |
> | Subject 7 | 0.8475 $\pm$ 0.0208   |
> | Subject 11| 0.4736 $\pm$ 0.0073   |
> | Subject 12| 0.5422 $\pm$ 0.2065   |
> | Subject 13| 0.4094 $\pm$ 0.0178   |
> | Subject 14| 0.7586 $\pm$ 0.0074   |
> | Subject 16| 0.5285 $\pm$ 0.0090   |
> | Subject 17| 0.6757 $\pm$ 0.0176   |
> | Subject 18| 0.6208 $\pm$ 0.0114   |
> | Subject 20| 0.6625 $\pm$ 0.0050   |
> | Subject 21| 0.5682 $\pm$ 0.0404   |
> | Subject 22| 0.7875 $\pm$ 0.0379   |
> | Subject 23| 0.7152 $\pm$ 0.0015   |
> | Subject 24| 0.5862 $\pm$ 0.0020   |
> | Subject 26| 0.4531 $\pm$ 0.0081   |
> | Summary   | 0.6344                |
>
> | ConvTran | MI                     | SSVEP                  | ERN                     |
> |----------|------------------------|------------------------|-------------------------|
> | NoSub    | 0.4796 $\pm$ 0.0073     | 0.2918 $\pm$ 0.0218    | 0.6994 $\pm$ 0.0074     |
> | Concat   | 0.4464 $\pm$ 0.0733     | 0.3009 $\pm$ 0.0091    | 0.6920 $\pm$ 0.0155     |
> | IdEmb    | 0.4946 $\pm$ 0.0197     | 0.2897 $\pm$ 0.0127    | 0.7207 $\pm$ 0.0046     |
> | Head     | 0.5158 $\pm$ 0.0000     | 0.2918 $\pm$ 0.0145    | 0.6936 $\pm$ 0.0120     |
>
> ```
> We will add a detailed analysis in the appendix of the paper and have adjusted our contributions to these results.
> While some TSC models are able to outperform the EEG-specific models, not all of them can capture the EEG-specific
> noise patterns. Our current hypothesis is that max pooling, which is used in the inception model, is helping with
> denoising, while models like TimesNet, ConvTran, and to some extent, ResNet are not able to capture the noise
> patterns. We will further investigate this hypothesis in the future and have updated our conclusion and outlook with
> these findings.
> ```

---

### Review · Reviewer_EjDn · 2024-08-27

**Summary Of Contributions:**

To tackle the problem of EEG classification, the authors make the argument that the data should be treated as a time-series which would inform modeling choices. Moreover, the authors investigate building a joint model that is able to be trained on all subjects simultaneously.

**Audience:**

Yes

**Claims And Evidence:**

No

**Requested Changes:**

- Clearly define what is the contribution of the paper (Crucial)
- Fix flow of the paper (Crucial)
- More experimental details (Crucial)

**Strengths And Weaknesses:**

# Strengths
I think the coolest thing about the paper is the joint training across subjects as not only is this practically useful (as training and maintaining one model is favorable to having a model per patient) but also theoretically interesting.

# Weaknesses
The biggest weakness of the paper is the presentation, as it is not necessarily clear what the goal nor the take-away is. The paper begins by stating that EEG classification should be viewed as a time-series classification problem and then states that they use ResNet and Inception for this problem. Here is where the problems start: first, the initial exposition implies that previous EEG works don't treat EEG data as time-series but in the related works section, most, if not all, of the previous approaches use temporal convolution (which means that they are indeed treating the data as a time-series). It is also odd that the authors use convolutional architectures as their baselines but not recurrent neural networks, which are a corner-stone for deep learning on time series.

Next, the authors spend too much space on the definition of EEG classification and time-series classification. Moreover, the notation used is confusing. For instance, the authors state the following on page 4 "By $\mathbb{R}^{* \times C} = (\mathbb{R}^C)^* = \underset{T \in \mathbb{N}}{\cup} \mathbb{R}^{T \times C}$ we denote the finite sequences of vectors with C dimensions." This is simply just $\mathbb{R}^{T \times C}$.

Arguably, the most interesting and novel part of the paper isn't discussed until section 4 which is designing and training a joint model. This section deserves the most space but sadly, isn't only given 3/4 of a page.

Lastly, from the experiments section, it is not clear what the takeaway is. It seems like the proposed approaches tend to significantly underperform opposed to the previous approaches. While this is fine, no time is spent discussing this which I think is paramount. As all the models seem to be performing some sort of temporal convolution, all of them are doing time-series classification. Thus, what is the takeaway; that joint training is difficult? Also, there are no details of the pre-processing of the data in the paper!

---

> ### Author Response · Authors · 2024-09-09
> **Response to Reviewer EjDn**
>
> We thank the reviewer for their detailed review and would like to address the individual points:
>
> **Weaknesses**
>
> EEG works don't treat EEG data as time-series
> ```
> We agree with the reviewer that dedicated EEG models use building blocks from time series. However, while they use these
> building blocks, baseline models from the time series literature are rarely used as baselines, and the EEG models are
> developed in a vacuum. We want to show that simple/common baselines are already very competitive out-of-the-box and
> should be considered. This was clarified in the introduction and related work section of our revision.
> ```
>
> Notation:
>
> The notation $R^{* \times C}$ is not simply $R^{T \times C}$ for a specific $T$ but the union of $R^{T \times C} \forall T \in N$, i.e.,
> we allow sequences of different lengths. The notation $R^{C^*} = R^{T \times C}$ only says, that we follow the usual convention of not differentiating between the sequences of vectors and matrices with suiting shapes. We have clarified this in our revision.
>
>
> Arguably, the most interesting and novel part of the paper isn't discussed until section 4
> ```
> We have reordered the sections to give more prominence to the joint training protocol in our paper and have also
> expanded our analysis of this protocol in the Experiment section.
> ```
>
> It seems like the proposed approaches tend to significantly underperform opposed to the previous approaches.
> ```
> The time series models, as well as our proposed methods for joint subject training, are unable to capture the noise
> patterns of the EEG data for the first MI dataset. One explanation for this is the longer sequence length compared to
> the other two datasets, where the patching of the sequence done by MAtt provides a clear advantage, and we have added
> this to the paper. For the other two datasets, the joint time series models outperform all other dedicated EEG baselines
> or are competitive (second best), as shown in Table 3. This supports our main argument that EEG literature should be
> compared against vanilla time series classification (TSC) models with subject embeddings. However, we also agree that
> our proposed solutions for the subject-conditional approaches are not yet perfect and require further research. A joint
> training approach might not be possible for all datasets at this point in time.
> ```
>
> Pre-processing
> ```
> Regarding the preprocessing, we mention that we adopt the same protocol as MAtt, and we have added the details to our
> paper as well.
> ```
>
> **Requested Changes**
>
> Clearly define what is the contribution of the paper (Crucial)
> ```
> We have updated our contributions to the following:
> 1. We argue that EEG-series sequences are a special form of time-series, namely time-series with one categorical
> attribute (the subject ID).
> This indicates, that they should be tackled with time-series models which can incoporate such additional attributes.
> 2. We provide a theoretical framework to classify EEG models into three categories:
> - subject-specific, where separate models are learned for each subject
> - subject-agnostic, where one joint model for all subjects is learned without subject information,
> - subject-conditional, where one joint model for all subjects is learned while utilizing subject
> information.
> 3. We propose three novel methodologies for subject-conditional EEG classification. Our procedure is model-agnostic and can
> be integrated into any differentiable classification model.
> 4. We show that some simple time-series classification baselines such as ResNet and Inception can be competitive or even
> outperform dedicated EEG classification models in 2 out of 3 cases, while other TSC models are unable to capture the EEG
> inherent noise pattern.
> ```
>
> Fix the flow of the paper (Crucial)
> ```
> We believe that our current flow aligns with our newly formulated contributions:
> First, we discuss time series with categorical attributes (Contribution 1, Section 3).
> Next, we address the theoretical joint training framework (Contribution 2, Section 3.4).
> Then, we present our subject-conditional approaches (Contribution 3, Section 4).
> Finally, we showcase our experimental results (Contribution 4, Section 6).
>
> We hope, that the flow of our paper is now clear to the reviewer. If not, please let us know, which additional changes
> are needed.
> ```
>
> More experimental details (Crucial)
> ```
> We have added the details for the preprocessing of the data as well as the additional analysis described above.
> Furthermore, we have particularly focused on the main findings regarding the joint-subject training and expanded the
> discussion of the analysis in the ablation study. Additionally, we have implemented TimesNet as well as ConvTran as an
> additional baseline. The results can be found in our response to reviewer RCq4 and we will provide a detail analysis in
> the appendix.
> ```

---

### Decision · Action_Editor_GFye · 2024-10-15

**Recommendation:** Reject

**Comment:**

The main feedback from multiple reviewers is that claims and definitions are unclear, making it challenging to be in the category of scholarship  with accurate, convincing and clear evidence.

**Audience:**

Yes.

**Claims And Evidence:**

The claims made by the paper are unclear for multiple authors. Additionally, it makes some claims that are wrong (1) that EEG is newly being treated as a time-series problem or that (2) deep learning models for EEG data will lead to better understood-learning on EEG data. Deep learning is powerful, but not for every single use-case. It should be introduced in a tactful manner.

**Resubmission Of Major Revision:**

The authors may consider submitting a major revision at a later time.